# Syndromic Surveillance of Health Effects Due to Summer Sun Overexposure: Construction of an Indicator Based on Drug Sales in Pharmacies—Preliminary Study within the PRISME Project

**DOI:** 10.3390/ijerph20136287

**Published:** 2023-07-03

**Authors:** Adeline Riondel, Leslie Simac, Olivier Catelinois, Carole Morlan-Salesse, Frédéric Bounoure, Bruno Galan, Damien Mouly

**Affiliations:** 1Santé Publique France, Regions Division, 34000 Montpellier, France; 2The Regional Health Agency of Occitanie, 34000 Montpellier, France; 3D2CN, UFR of Health, NORDIC Laboratory, INSERM 1239, University of Rouen, 76128 Mont Saint Aignan, France; frederic.bounoure@hopital-yvetot.fr; 4Regional Council of the Order of Pharmacists (CROP) of Occitanie, 34000 Montpellier, France

**Keywords:** sunburn, solar UV, health impact, health care use, health care consumption, pharmacy, medicines

## Abstract

Introduction: Solar radiation is classified as a known human carcinogen. In France, people frequently ask local pharmacies to dispense products for sunburns. In the PRISME project, studying this use can be a specific and sensitive way to assess these overexposures. Objective: This study aims to construct an indicator for monitoring healthcare consumption in pharmacies after overexposure to solar UV. Methods: The study, conducted between July and August 2019, covered a sample of pharmacies located in coastal communities of southern France. A list of products for sunburn was defined. When one of the products on this list was sold, the customer was asked to fill out a questionnaire to determine whether the purchase was related to UV overexposure. A positive predictive value (PPV) per active ingredient was calculated. Results: Overall, nine pharmacies participated in the study, and 288 questionnaires were collected. The majority of products purchased were for women (60.7%), for people aged 15 and over (78.1%), and for people not living in the department (68.9%). The most frequently purchased products were our trolamine-containing products which accounted for 53% of sales. With the exception of three products, all PPVs were greater than 0.8. Conclusion: The high PPV confirms the suitability of the product selection as an indicator for monitoring healthcare consumption related to solar UV overexposure. Two indicators (one sensitive and one specific) were selected to maximise the chances of identifying UV-related remedies.

## 1. Introduction

With almost 80,000 new cases per year in France, skin cancer (basal cell carcinoma, squamous cell carcinoma and melanoma) is one of the most common cancers, with a substantial increase in incidence occurring over the last 50 years [1]. The most serious form, cutaneous melanoma, has seen a sharp increase in incidence since the 1980s [2,3]. According to the International Agency for Research on Cancer (IARC), nearly 70% of cutaneous melanomas are caused by sun exposure [4], and this exposure affects the entire population. The demonstrated carcinogenic effects of UV (UVA and UVB) have led the IARC to classify solar radiation as a definite human carcinogen (Group 1) [5]. In addition, prolonged and too intense exposure to UV radiation can lead to short-term risks, such as cutaneous burns, weakening of the immune system or heat stroke; medium-term risks, such as photo keratitis, photo conjunctivitis and premature ageing of the skin; and long-term risks, such as early cataracts [6].

The short-term effects of UV overexposure can be dermatological (sunburn, also known as solar erythema, first- or second-degree burns of varying size and different locations with or without phlyctens) and/or ophthalmological (ophthalmia or photo keratoconjunctivitis). Depending on the severity of the condition, treatment can be provided by pharmacies, outpatient clinics or hospital emergency departments.

Since 2003, Santé publique France has developed a syndromic health surveillance system based on the collection of non-specific medico-administrative data. It allows the daily centralisation of information from emergency structures participating in the coordinated emergency surveillance network (OSCOUR^®^), from SOS Médecins associations (data from town emergency medicine) and from municipalities for mortality data via the National Institute for Statistics and Economic Studies (INSEE) (https://www.santepubliquefrance.fr/surveillance-syndromique-sursaud-R (accessed on 13 February 2023)). Establishments with emergency facilities see their overall activity strongly increase each summer, particularly those located near the coast. Analysis of the data shows that every year there is an increase in the number of emergency room visits for burns on the southern French coast from May to August, corresponding to the tourist influx [7]. In supervised bathing areas on beaches, lifeguards can direct beachgoers with minor injuries to pharmacists, who are both more accessible and geographically closer to the beaches than other health services. Given that pharmacists must contribute to informing and educating the public on health matters (French Public Health and Ethics Code, article R4235-2 (https://www.legifrance.gouv.fr/codes/article_lc/LEGIARTI000006913652 (accessed on 25 May 2023))) and as they are obligated to provide advice when dispensing medication that does not require a medical prescription (article R4235-48 (https://www.legifrance.gouv.fr/codes/article_lc/LEGIARTI000006913703 (accessed on 25 May 2023))), they are regularly called upon to deal with the short-term effects of minor cases of overexposure to the sun.

There are very few studies on the use of healthcare to treat health effects related to solar overexposure. The literature has focused more on the use of emergency facilities or equivalents [8,9,10,11,12]. One of these studies, carried out in France [13], analyses the reason for emergency room visits by foreign tourists but does not look at the referral to care, i.e., the place of referral or the type of care system sought. Few precise data were found on the health impact of overexposure to solar UV and the effect on pharmaceutical consumption. Nevertheless, in a US study analysing the incidence of sunburn in a cohort of 75,614 beachgoers, it was described that, of those who experienced sunburn (9882 or 13.1%), 34% used self-medication from US pharmacies (OTC: over-the-counter consumption) [14].

The use of local pharmacies, therefore, appears to be frequent and results in the consumption of many recommended or prescribed specialities. This complementary data source provides information on the less-severe effects of solar exposures with a greater level of geographical precision compared to emergency data (emergency departments or SOS Médecins associations). This geographical precision can be useful for future studies on the link between solar exposures and meteorological UV indicators. Evaluating these uses can then make it possible to establish a specific indicator of the short-term effects of overexposure to solar UV.

This preliminary study is part of the impact component of the PRISME project (Prevention and Impact of Solar Exposure on the Mediterranean Coast), which aims, on the one hand, to orientate actions to reduce the health impact of solar exposure by adapting prevention campaigns on the basis of current knowledge and behaviours (prevention component) [15] and, on the other hand, to better describe the health impact of short-term effects (impact component).

Indeed, the study region in the PRISME project, because of its location in the south of France, its four coastal departments, 32 towns, and its Mediterranean coastal strip of more than 200 km, is a region particularly concerned by solar exposure (Geographic coordinates for le Grau-du-Roi, northernmost town are 43°32′17″ N, 4°08′14″ E; for Cerbère, the southernmost town is 42°26′39″ N, 3°09′56″ E). This exposure concerns the resident population of the coast (1.3 million permanent inhabitants), as well as the tourist population (about 8 million per year) (https://www.laregion.fr/L-avenir-maritime-d-Occitanie-se-construit-des-aujourd-hui-avec (accessed on 13 February 2023)), which is very significant during the summer period in the 20 seaside resorts of the coast and is subject to intermittent and intense exposure to solar UV.

The objective of this study is to construct an indicator of the dermatological effects of solar overexposure for the resident and tourist populations based on the consumption of pharmaceutical products in pharmacies and to evaluate the performance of this indicator.

## 2. Methods

### 2.1. Pharmacy Selection

This prospective observational study took place between July and August 2019 in 73 pharmacies located in a subselection of 15 coastal towns of the PRISME project area (Mediterranean coast, southern France). Pharmacies in the direct vicinity of the coast were favoured, especially the larger ones, which are likely to have an increase in activity during the summer in connection with the influx of tourists. The participating pharmacies were identified and selected in collaboration with the Conseil Régional de l’Ordre des Pharmaciens (CROP). Each of them was visited to explain the purpose of the study, to ask them to participate and, if necessary, to distribute the patient questionnaires. Pharmacy participation was voluntary, unpaid, and no additional human resources were provided to help fill out the questionnaires. Hence, they filled out as many questionnaires as they could during this period.

### 2.2. Product Selection

A first list was established with products based on trolamine, an active ingredient found in European and American pharmacopoeias, that could be prescribed to treat the dermatological effects of overexposure to the sun. As there were few CIPs (Code Identification of the Presentation for a product) and as there were no systematic prescriptions by a general practitioner to treat these effects, the list was expanded to include products often sold by pharmacists over the counter within their usual sales practices, based on discussions with expert pharmacists, the president of the CROP and a pharmacist from the Regional Health Agency. This list of varied products made it possible to cover as many cases as possible, regardless of their degrees of severity.

### 2.3. Inclusion/Exclusion Selection Criteria for Participants

The effects of the sun considered in this preliminary study were dermatological effects: minor superficial burns, erythema, non-infected superficial wounds and skin irritations. A questionnaire had to be filled out directly with the patient whenever at least one of the pharmaceutical products on the target CIP list was sold. The same procedure was followed for patients presenting at the pharmacy for advice or to buy other products linked to such sun-related dermatological effects. It was specified to the pharmacist that a CIP not on the list could be added to the questionnaire to improve the proposed list. Sales related to dermatological ailments without a link to the sun or non-dermatological effects linked to the sun were not included

### 2.4. Data Collection

The study period during which each pharmacy could participate ranged from one week to more than six weeks, depending on the ability of the pharmacies to absorb the additional workload. The weeks of participation were different from one pharmacy to another in order to avoid possible climate-related hazards (variations in sunshine and/or temperatures), which could have repercussions on the use of care for short-term effects.

No data collected could directly or indirectly identify a person. For this reason, no individual consent was needed. The questionnaire (Figure 1) was filled out by the pharmacist directly with the person purchasing the product(s). The following information was requested for each purchase: the age of the person for whom the product was intended (by age group), the notion of direct purchase (after advice from the pharmacist or not) or on prescription, if the purchase/prescription was in connection with sun exposure, and if the person usually resided in the department. About ten questionnaires per day per pharmacy were considered a reasonable completion target; exhaustivity was not required.

### 2.5. Bias

Means were put in place to limit selection bias before and during the study: pharmacies were recruited directly through on-site visits with explanatory documents (questionnaire, patient information letter), and a member of the team was trained to administer the standardised questionnaire and select the person to be interviewed. Several telephone reminders were also issued to pharmacies at the beginning and during the study period to remind them of the objectives and collection procedures. The initial study period was adapted for some pharmacies to suit their workload. Pharmacies further away from the coast were recruited in order to have a greater diversity of respondents.

### 2.6. Statistical Analyses

A descriptive analysis of the frequency of purchase of products was carried out.

The positive predictive value (PPV), equivalent to the number of true positives (sales linked to overexposure to the sun) divided by the cumulative number of true and false positives (all the sales), was also calculated. True positives (TPs) corresponded to pharmaceutical products purchased in the context of UV overexposure on the basis of the patient’s declaration in the questionnaire. False positives (FPs) corresponded to specialities on the target CIP list purchased to treat another condition unrelated to UV exposure.

The active ingredients have been grouped according to their ATC class (anatomical, therapeutic and chemical) (https://assurance-maladie.ameli.fr/etudes-et-donnees/medicaments-classe-atc-medicam (accessed on 13 February 2023)). Combined PPVs were then calculated for products purchased simultaneously and ATC class groupings in order to assess differences between PPVs of different ATC classes. The PPV of all CIPs was also calculated.

## 3. Results

### 3.1. Pharmacy Recruitment and Data Collection

The recruitment of pharmacies led to the participation of 14 pharmacies out of the 73 in the study area, i.e., 19% of the pharmacies open. The area of selection of participating pharmacies covers a part of the Mediterranean coastline, with 11 pharmacies located within 5 km of the coastline and 3 pharmacies located between 5 and 15 km from the coastline. The period of participation in the collection of information from the different pharmacies ranged from 1 to 6 weeks.

At the end of the study, it was found that five pharmacies had not completed any questionnaires, mainly due to lack of time and being too busy during the period, reducing the number of participating pharmacies to nine (64% participation rate). A total of 288 questionnaires were collected during the summer of 2019. Pharmacies 1 and 13 collected respectively 25.3% and 30.6% of the questionnaires (the seven other participating pharmacies collected 3 to 12% each, for a total of 44.1%). The number of completed questionnaires per follow-up day was less than one in five out of nine pharmacies. Pharmacies 1, 8, 10 and 13 completed questionnaires fairly regularly (more than 75% of the follow-up days with at least one completed questionnaire).

### 3.2. Study Population and Description of Products Purchased

The majority of the products purchased were for women (60.7%) and people aged 15 years and over (78.1%). The individuals interviewed did not reside in the department in 68.9% of cases. In total, 280 products were purchased for 277 completed questionnaires. No product was ticked or added for 11 questionnaires: these consultations in the pharmacies were linked to overexposure to the sun, maybe based on advice (there was no additional information provided). The majority of purchases were for a single product (92.2%). Only two questionnaires were completed following the purchase of two products (Biafine Act ^®^ (186 g) and Biafine Act^®^ (139.5 g)) and three products (Biafine^®^ (93 g), Cicaderma^®^ and Osmosoft^®^). The most common were Osmosoft^®^ (38.2%), Biafine^®^ (32.1%) and Biafine Act^®^ (17.9%), i.e., 50% Biafine^®^. The remaining purchases are detailed in Table 1. The “Other” category was Dexeryl Specific Burns and Sunburn (nine products) and Tryptine Sunburn lotion (one product). Note that ATC class D03AX12 containing the same active ingredient, trolamine, comprising Biafine, Biafine Act, Lamiderm and Trolamine, represented 53% of the products purchased (150 products in total).

The majority of purchases were reported to be related to overexposure to the sun (84.9%). The majority of products were purchased without prior advice from the pharmacist (62.5%), following the pharmacist’s advice in 36% of cases and on medical prescription in 2.1% of cases. The six products delivered following a medical prescription were Osmosoft^®^ (*N* = 2), Biafine^®^ (*N* = 2), Biafine Act^®^ (*N* = 1) and Calendoron^®^ (*N* = 1). Of these six products, three were prescribed in connection with sun overexposure in people aged 15 and over (two Osmosoft^®^ and one Biafine^®^), and three were not: one Biafine Act^®^ for a child under 5 years of age and one Calendoron^®^ and one Biafine^®^ for persons aged 15 years and over. Osmosoft^®^ was mostly purchased on the advice of the pharmacist (54% of cases), while Biafine^®^ was mostly purchased directly without prior advice from the pharmacist (83%).

### 3.3. Positive Predictive Values

A total of 275 products were purchased on their own, of which 267 were found to have a “link to overexposure to the sun”. Most of the PPVs are greater than 0.8, except for Calendoron^®^ (PPV = 0), Cicaderma^®^ (PPV = 0.56) and Lamiderm^®^ (PPV = 0.33) (Table 2).

Grouping the products according to their ATC class, we note that the PPV of class D03AX12 (Biafine^®^, Biafine Act^®^, Lamiderm^®^, Trolamine) is equal to 0.80, while that of class D02AX, including Cicaderma^®^ and Calendoron^®^, is 0.50. It should be noted that Osmosoft^®^ and Urgo Sunburn do not fit into the ATC classification because they are medical devices, even though Osmosoft^®^ was one of the most frequently sold products.

By grouping the seven pharmacies that filled 43.0% and comparing them with pharmacies 1 and 13 (which collected 25.3% and 30.6% of the data, respectively), the PPVs remain comparable (between 0.71 and 0.91 for Biafine^®^, 0.81 and 1 for Biafine Act^®^, 0.88 and 1 for Osmosoft^®^). Nevertheless, Cicaderma^®^ has a PPV of 0.33 in pharmacy 13 compared to a PPV of 1 when grouping the other pharmacies.

### 3.4. Indicator

The PPV analysis allowed the list of target CIPs to be adjusted for the construction of an indicator to measure the short-term effects of solar UV overexposure.

It was decided to construct two indicators: indicator A, focusing on specificity, and indicator B, focusing on sensitivity. The indicators are defined by the aggregated sum of the number of products sold daily per pharmacy from a list of defined products.

To construct indicator A, we selected the CIPs with PPVs greater than 0.7, considering that these codes were used in the context of UV overexposure (no literature on the subject, threshold set arbitrarily). We then added CIPs belonging to the same ATC class as a product with a PPV greater than 0.7.

To construct indicator B, because of the small numbers for some codes, product indications were taken into account to increase sensitivity.

Because the collection of products was not included initially, the indicator including CIPs associated with overexposure to the sun was adjusted as follows: addition to the CIP list of Urgo Emulsion Sunburn and Burns, Tryptine Sunburn and Dexeryl Specific. These evolutions are linked with products added in questionnaires by the participating pharmacists and because of some new products according to market development.

Indicator A is composed of the daily count per pharmacy of sales of the following products: Agathol^®^, Biafine^®^, Biafine Act^®^, Brulex^®^, Dexeryl specific, Lamiderm^®^, Osmosoft^®^, Trolamine, Tryptine Sunburn, Urgo Emulsion Sunburn and Burns, Urgo Sunburn Spray (Table 3). This indicator is associated with a PPV per product greater than 0.7, except for Lamiderm^®^ added according to the ATC class criteria. Indeed, Lamiderm^®^ has a low PPV but is in the same ATC class as Biafine^®^ and Trolamine with PPVs above 0.7.

In order to not omit an undocumented link in our study between a CIP and sun overexposure, we constructed the joint indicator B, grouping all previously selected CIPs, to increase sensitivity, i.e., the ability to identify effects related to sun overexposure. This indicator is associated with a PPV per product greater than 0.3.

The list of CIPs included in the two indicators changes to be more or less sensitive. It may also be adapted in line with market developments.

## 4. Discussion

### 4.1. Strengths of the Study

The study collected data on the activity of pharmacies during the summer period regarding the sale of products used for the treatment of sun overexposure and obtained the demographic characteristics of consumers.

The majority of products were purchased by people living outside the study area, a trend consistent with the region’s summer tourist activity.

A higher proportion of women and people aged 15 and over bought these products. This can be explained by the fact that the under-15s are more protected from the sun by their parents or that they are less exposed to the sun than teenagers/adults (impact of prevention messages), or by a reporting bias (purchase declared for an adult when this is not the case). Women may be more likely to be in charge of health expenditure and care within the household [16] and also more aware of prevention messages compared to men [17].

The study also made it possible to construct an indicator based on the sales of specialities in pharmacies in order to measure the short-term impact of solar overexposure on people’s health and to evaluate the PPV.

### 4.2. Limitations and Biases

The main limitation of our study is the low participation of pharmacies in the study, mainly due to lack of time, as those located on the seaside are extremely busy during the summer period. These pharmacies are also the most likely to be in contact with tourists or people with overexposure to UV.

The collection of sales of the products listed in the questionnaire was not exhaustive. This can be explained by the lack of time. If such a study were to be conducted again, dedicated means (online questionnaires or additional human resources) would certainly increase the collected information (number of questionnaires). However, the quality of data collected is not discussed. Even though our aim was not to reach the exhaustiveness of sales, this partial collection may lead to a selection bias of respondents.

Despite possible selection biases, our study is consistent with a study carried out at the same time in another region of the French coast (North Atlantic coast) [18]. This study applied the same protocol and questionnaire as our study. Thirty-five questionnaires were completed, and despite the small number of respondents, the main results are consistent with those of our study. The main products purchased were the same: Osmosoft^®^ (14%), Biafine^®^ (17%) and Biafine Act^®^ (34%). The overall PPV was 71.4% for this study from the North Atlantic coast. This value is below the PPVs of 81% to 94% found for the main products in our study in the south of France but can be explained by a weather-induced reduction in the proportion of sunburn.

The CIPs selected for the study only concern UV-related dermatological effects. CIPs specifically targeting UV-related ophthalmological effects were not identified, probably because the symptoms are less specific and are not managed in pharmacies in relation to the severity of the symptoms (retinopathy or maculopathy, for example).

Finally, our study allowed us to detect recourse to pharmacies for solar erythema, which constitutes a proportion not known in France. It is 34% according to the American study [14]. This proportion can be constant over time.

### 4.3. Construction of the Indicator

The PPV of each product was estimated on the sample of available data. The results confirm that the pre-identified CIPs are related to sun exposure, except for some products (Cicaderma^®^, Lamiderm^®^, Calendoron^®^).

Some PPVs should be interpreted with caution because of the small numbers in those categories (e.g., Calendoron^®^, Lamiderm^®^, Trolamine, Cicaderma^®^), and no data were collected for Agathol^®^, Brulex^®^ and Dexatopia^®^. Cicaderma^®^ had a PPV of 0.33 in pharmacy 13 compared to a PPV of 1 when the other pharmacies were combined. Because of the small number of products sold, the exclusion of a pharmacy strongly affects the PPV of that product. The indications for the products, in addition to the PPVs, were, therefore, taken into account in order to construct a more sensitive indicator (indicator B).

Some products have a less specific indication for sun-related effects (which may explain the lower PPV of Calendoron^®^) and have been excluded from indicator A because of the background noise they may cause. These are Calendoron^®^ (homoeopathic treatment of minor wounds and burns, as well as skin irritations), Cicaderma^®^ (homoeopathic treatment of minor wounds, burns and insect bites) and Dexatopia^®^ (adjuvant treatment of skin dryness and minor superficial burns)

Biafine^®^, Biafine Act^®^ and Osmosoft^®^ were the most frequently purchased products in this study. These products were offered in large quantities and displayed in most of the pharmacies we recruited (dedicated shelves or displays). They are also very specific in their indications and have been included in indicators A and B.

Biafine Act^®^ has a higher PPV of effects related to sun overexposure than Biafine^®^, which can be explained by its more specific indication for solar erythema (small superficial burns, including localised sunburn and non-infected wounds), whereas Biafine^®^ can be indicated for all types of burns (treatment of burns, non-infected superficial wounds and post-radiation redness).

Despite the low PPV for Lamiderm^®^, this CIP code was included in our indicators A and B. Lamiderm^®^ and Trolamine have the same indications as Biafine and belong to the same ATC class.

Agathol^®^ and Brulex^®^ have the same indication, i.e., adjuvant treatment of small superficial burns. We did not collect data for these two products, but the CIP codes were kept in the indicators in order to compare the evolution of sales over the seasons and to see if there could be a link with the UV index.

Finally, some products are described with few data but are nevertheless very specific. This is the case for Urgo Emulsion Sunburn and Burns and Tryptine Sunburn, which we have kept in indicators A and B. Similarly, Dexeryl Specific Burns and Sunburn was not on the market when the questionnaire was drawn up, but as it is very specific, it should be included in the two indicators. Note that these products do not have a CIP as they are medical devices.

The list of products is subject to change according to new specialities that may arrive on the market, the withdrawal of specialities or packaging, and new treatments.

## 5. Conclusions and Outlook

This exploratory study made it possible to define a specific indicator A concerning the consumption of treatments for solar overexposure in coastal pharmacies. This indicator seems reliable, but the small amount of data collected and the lack of literature make it difficult to select a definitive list of CIP codes. An indicator B, grouping all the CIP codes studied, was also proposed to increase sensitivity.

The study should continue by collecting sales data for the identified products by direct and automatic extraction from pharmacy databases. Because of the high number of people visiting these pharmacies during the summer period, this will maximise the possibility of capturing the healthcare consumption of both tourists and residents. They will make it possible to contribute to quantifying the use of healthcare by the population in relation to overexposure to solar UV. To do this, they will have to be completed with data from medical emergencies, which are also the subject of work in progress.

## Figures and Tables

**Figure 1 ijerph-20-06287-f001:**
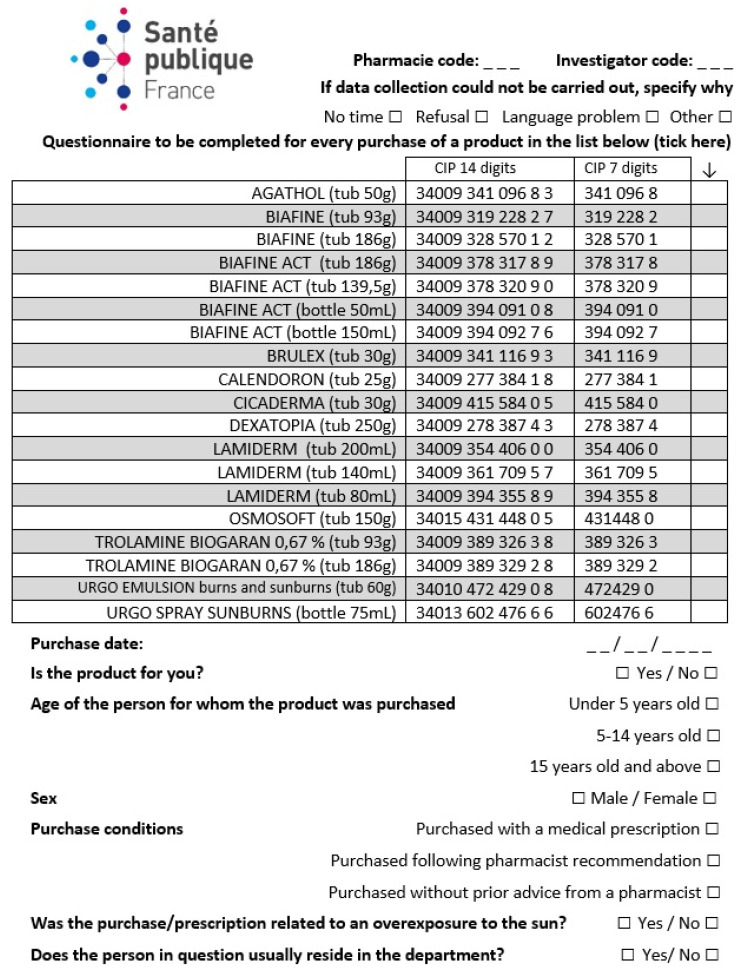
Questionnaire filled out by the pharmacist interviewing the person who was buying the product.

**Table 1 ijerph-20-06287-t001:** Demographics and products purchased in pharmacies during the study in summer 2019.

	MD	*N*	Adjusted
Age of the person for whom the product is intended	32		
15 years and over		200	78.1
5–14 years		44	17.2
Under 5 years		12	4.7
Sex	6		
Female		170	60.7
Male		98	35.0
Male + female		14	5.0
Person residing in the department	8		
No		193	68.9
Yes		87	31.1
Product	11		
Osmosoft^®^		107	38.2
Biafine^®^		90	32.1
Biafine Act^®^		50	17.9
Cicaderma^®^		10	3.6
Other		10	3.6
Lamiderm^®^		6	2.1
Trolamine Biogaran 0.67%		4	1.4
Urgo Sunburn Spray		2	0.7
Calendoron^®^		1	0.4
Urgo Emulsion Sunburn and Burns		0	0.0
Brulex^®^		0	0.0
Agathol^®^		0	0.0
Dexatopia^®^		0	0.0

MD = Missing data.

**Table 2 ijerph-20-06287-t002:** Positive predictive value by product and ATC class and link to sun overexposure in summer 2019.

	Link to Overexposure to the Sun			
	MD	Yes	No	PPV
Product name				
Osmosoft^®^ *	1	99	6	0.94
Biafine^®^	1	71	17	0.81
Biafine Act^®^	5	36	7	0.84
Cicaderma^®^	-	5	4	0.56
Lamiderm^®^	-	2	4	0.33
Trolamine	-	4	0	1.00
Urgo sunburn *	-	2	0	1.00
Calendoron^®^	-	0	1	-
Agathol^®^	-	0	0	-
Brulex^®^	-	0	0	-
Dexatopia^®^	-	0	0	-
ATC Class/Active Ingredient				
D02AB/zinc oxide (Agathol^®^, Brulex^®^)				
D03AX12/Trolamine (Biafine^®^, Biafine Act^®^, Lamiderm^®^, Trolamine)	6	113	28	0.80
D02AX/Calendula officinalis tincture (Calendoron^®^, Cicaderma^®^)	-	5	5	0.50
D02AC/Glycerol, petrolatum, liquid paraffin (Dexatopia^®^)	-	0	0	-

MD = Missing data. * Medical devices, not medicines and therefore not covered by an ATC class. D02AB: dermatologicals (D), emollients and protectives (02), emollients and protectives (A), zinc oxide–based medicine (B). D03AX12: dermatologicals (D), preparations for treatment of wounds and ulcers (03), cicatrizant (A), other cicatrizants (X), trolamine. D02AX: dermatologicals (D), emollients and protectives (02), emollients and protectives (A), other emollients and protectives (X). D02AC: dermatologicals (D), emollients and protectives (02), emollients and protectives (A), paraffin and fatty products © (D).

**Table 3 ijerph-20-06287-t003:** Products included in the indicator according to positive predictive value.

Product Name	CIP or EAN * Code	Active Ingredient	Indicator A	Indicator B
AGATHOL^®^ (50 g tube)	34009 341 096 8 3	Zinc oxide, titanium dioxide, Peru balsam	X	X
BIAFINE^®^ (93 g tube)	34009 319 228 2 7	Trolamine	X	X
BIAFINE^®^ (186 g tube)	34009 328 570 1 2	Trolamine	X	X
BIAFINE ACT^®^ (186 g tube)	34009 378 317 8 9	Trolamine	X	X
BIAFINE ACT^®^ (139.5 g tube)	34009 378 320 9 0	Trolamine	X	X
BIAFINE ACT^®^ (50 mL bottle)	34009 394 091 0 8	Trolamine	X	X
BIAFINE ACT^®^ (150 mL bottle)	34009 394 092 7 6	Trolamine	X	X
BRULEX^®^ (30 g tube)	34009 341 116 9 3	Zinc oxide, phenazon, phenol, sodium salicylate peru balsam	X	X
CALENDORON^®^ (25 g tube)	34009 277 384 1 8	*Calendula officinalis* tincture		X
CICADERMA^®^ (30 g tube)	34009 415 584 0 5	*Calendula officinalis* flowering top, *Hypericum perforatum* flowering top, *Achillea millefollium* flowering top, *Ledum palustre* mother tincture		X
DEXATOPIA^®^ (250 g tube)	34009 278 387 4 3	Glycerol, petrolatum, liquid paraffin		X
DEXERYL SPECIFIC BURNS AND SUNBURN ** (150 g tube)	35770 560 189 2 3	-	X	X
LAMIDERM^®^ (200 mL tube)	34009 354 406 0 0	Trolamine	X	X
LAMIDERM^®^ (140 mL tube)	34009 361 709 5 7	Trolamine	X	X
LAMIDERM^®^ (80 mL tube)	34009 394 355 8 9	Trolamine	X	X
OSMOSOFT^®^ (150 g tube)	34015 431 448 0 5		X	X
TROLAMINE BIOGARAN 0.67% (93 g tube)	34009 389 326 3 8	Trolamine	X	X
TROLAMINE BIOGARAN 0.67% (186 g tube)	34009 389 329 2 8	Trolamine	X	X
TRYPTINE SUNBURN ** (150 mL bottle)	34015 657 464 7 6	-	X	X
URGO EMULSION sunburn and burns ** (60 g tube)	34010 472 429 0 8	-	X	X
URGO SUNBURN SPRAY ** (75 mL bottle)	34013 602 476 6 6	-	X	X

* EAN code associated with medical devices. ** Medical devices, not medicines and therefore not concerned with an active ingredient. The composition of each product is available on https://www.vidal.fr/parapharmacie.html (accessed on 25 May 2023).

## Data Availability

The data presented in this study are available on request from the corresponding author, but they are not publicly available because of their link with the commercial activity of each pharmacy.

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
