# Peer review of "Syndromic Surveillance of Health Effects Due to Summer Sun Overexposure: Construction of an Indicator Based on Drug Sales in Pharmacies—Preliminary Study within the PRISME Project"

_ijerph, 2023, doi:10.3390/ijerph20136287_

Round 1

Reviewer 1 Report

See attached review

Author Response

Dear reviewer,

Thank you for your comments. It make it possible to correct and to enhance our article. You can find our answers in the attached document.

Reviewer 2 Report

1.     What is the sample size calculations? Please state. What type of sampling ? Any inclusion or exclusion criteria? Please state in manuscript.

From results it states 14 out of 129 pharmacies, 11% seems too little to represent the region of studies stated. 5 pharmacies didn’t complete the questionnaires, hence results were from only 9 pharmacies out of 129.

2.     From results as well, 288 questionnaires were obtained. Again what is the sample size for the recruitment of participants as stated? Inclusion and exclusion criteria etc. Need to have more details on the methodology.

3.     How was the score calculated? Were the questionnaire validated before use for example using the Cronbach alpha? Please state in manuscript.

4.     Ethics approval for the study was not included? Please include in the manuscript.

5.     “link to overexposure to the sun" – how was this decided? What are the criteria used for predicting/deciding that its and overexposure to the sun. Need to give more details.

6.     How was the CIP decided? It was stated that expert pharmacists decided them but on what terms were these products chosen? The active ingredients ranges from physical, chemical sunscreen and natural products.

7.     “The main limitation of our study is the low participation of pharmacies in the study, 271 mainly due to lack of time, as those located on the seaside are extremely busy during the 272 summer period. These pharmacies are also the most likely to be in contact with tourists or 273 people with overexposure to UV” : lack of time is not a reason. Author should ensure proper data collected for the research to be accepted and not questioned, albeit if it takes longer time to obtained them.

8.     Comparison to other studies done elsewhere or related studies should be done. What is the significance to this findings and others? The products available in this region may not be available elsewhere. More references are required for further discussions.

Author Response

(The authors gave the same response as above.)

Round 2

Reviewer 1 Report

The authors have thoroughly addressed my comments.  The additional information will very likely be helpful to readers, particularly from countries where pharmacists do not play as big a role in addressing skin questions.